# Improving the Wear Life of a-C:H Film in High Vacuum by Self-Assembled Reduced Graphene Oxide Layers

**DOI:** 10.3390/nano9121733

**Published:** 2019-12-05

**Authors:** Hui Song, Gang Chen, Jie Chen, Hongxuan Li, Li Ji, Nan Jiang

**Affiliations:** 1Key Laboratory of Marine Materials and Related Technologies, CAS, Zhejiang Key Laboratory of Marine Materials and Protective Technologies, Ningbo Institute of Materials Technology and Engineering, Chinese Academy of Sciences, Ningbo 315201, China; songhui@nimte.ac.cn; 2Inner Mongolia Metallic Materials Research Institute, Ningbo 315103, China; cg661231@163.com; 3State Key Laboratory of Solid Lubrication, Lanzhou Institute of Chemical Physics, Chinese Academy of Sciences, Lanzhou 730000, China; lihx@licp.cas.cn (H.L.); jili@licp.cas.cn (L.J.)

**Keywords:** self-assembly, reduced graphene oxide, a-C:H film, wear, vacuum, carbon materials

## Abstract

As an energy-efficient surface modification method, self-assembly has been the subject of extensive research. However, its application on carbon film has been rarely reported. In the present work, a novel self-assembled reduced graphene oxide (RGO) was prepared on a-C:H film by a controllable self-assembly method, and the friction behavior of the RGO/a-C:H film was investigated under vacuum environment. Interestingly, the RGO/a-C:H film exhibited significant improvement of anti-wear ability in vacuum conditions under a high applied load of 5 N. As expected, the synergy lubrication effect of the RGO layer and a-C:H film should account for the excellent friction reduction and anti-wear ability of a RGO/a-C:H multilayer film.

## 1. Introduction

With mankind’s increased exploration into space, there are new challenges to achieving a low friction coefficient, satisfactory anti-wear ability, and high reliability of lubrication materials in vacuum environments. With high hardness, good chemical inertness, and ultralow friction coefficient, hydrogenated amorphous carbon (a-C:H) film is currently referenced as a viable candidate for lubricating materials in a space environment [1]. Nonetheless, the severe adhesive wear between a-C:H film and its counterpart materials inevitably occurs in vacuum, which greatly limits its wear life in space conditions. Graphene, as a two-dimensional form of crystalline carbon, is considered to be a potential solid lubricant due to its lamellar structure, good flexibility, outstanding Young’s modulus (1 TPa), and hardness. Ali Erdemir et al. at the Argonne National Laboratory reported that both the wear and friction of a steel surface could be reduced by some graphene layers in humid air and dry nitrogen environments [2]. In our previous study, the low friction of graphene layers at the engineering scale was also observed under vacuum [3]. Driven by this aspect, we designed a novel sputtering method to form graphene-like nanoclusters into amorphous carbon films, which endowed a-C:H film with outstanding elasticity, high hardness, and excellent tribological properties [4]. However, the direct deposition of graphene layers on a-C:H film and their tribological performance under vacuum has rarely been reported. 

Self-assembly as a facile, convenient, and less costly surface treatment technology can spontaneously chemisorb molecules on various substrate surfaces. Many studies disclose that effective graphene lubricating films can be constructed by the self-assembly method [5]. After oxidization, graphene is well covalently assembled onto substrates through proper coupling agents such as 3-aminopropyltriethoxysilane (APTES) self-assembled monolayers [6]. Thus, in the present study, reduced graphene oxide (RGO) layers were assembled on a-C:H film by employing a self-assembled APTES monolayer as the intermediate coupling agent. Afterward, the friction behavior and wear mechanism of the as-prepared RGO/a-C:H film were systematically studied in a vacuum environment. 

## 2. Experimental Section

Graphite oxide was obtained from graphite flakes (Sinopharm Chemical Reagent Co. Ltd., Shanghai, China) by using a modified Hummers method and the details of the preparation procedures are given in [7]. a-C:H film with a thickness of about 1.5 µm was prepared on polished stainless steel (1Cr18Ni9Ti) substrates by a medium frequency unbalanced magnetron sputtering method. The sputtering equipment and preparation process of a-C:H film have been described in our previous reports [4]. Prior to assembly, a-C:H film was cleaned and hydroxylated by piranha solution at the temperature of 90 °C for 30 min. Afterward, the hydroxylated a-C:H film was steeped in APTES solution for 30 min. Subsequently, a-C:H film with APTES was immersed in a GO aqueous solution at 80 °C for 12 h. The prepared product was a GO/APTES/a-C:H multilayer film, which was called the GO/a-C:H film for short. The GO/a-C:H film was then thermally reduced at the temperature of 200 °C for 2 h in an argon atmosphere, and the final obtained product was called the RGO/a-C:H film. Repeating the above process could assemble different RGO layers on a-C:H film, and these were called RGO(x)/a-C:H (x = 1, 3, 5, 7 and 10) films. The preparation procedure of the RGO/a-C:H film is described in Figure 1.

A field emission scanning electron microscope (FESEM, JEOL JSM-6701F, Tokyo, Japan) was applied to observe the surface morphology of the as-prepared films. The detailed bonding structures and phase compositions were characterized by the Jobin-Yvon HR-800 Raman spectrometer (HORIBA, Ltd, Paris, France) and D8 Advance x-ray diffractometer (Bruker Corporatio, Germany) with Cu *Kα* radiation (λ = 0.1542 nm). Tribological performance was investigated on a ball-on-disk tribometer in a closed vacuum chamber [6]. The GCr15 bearing steel ball (HRC 58-60) with a diameter of 6 mm was selected as the friction counterpart. The sliding speed was 300 r/min, the applied load was 5 N, and the pressure of vacuum chamber was about 1 × 10^−4^ Pa. If the value of the friction coefficient exceeded 0.3, the lubrication ability of the film was considered to be ineffective and the friction test was manually stopped. Each friction test was conducted at least three times under same experimental conditions to avoid errors.

## 3. Results and Discussion 

### 3.1. Raman and X-Ray Diffraction Characterization

To gain more insight into the structure and composition information, Raman and X-ray diffration (XRD) characterizations were performed to analyze the as-prepared RGO/a-C:H films. The Raman spectrum of the RGO/a-C:H films with different layers is shown in Figure 2a. Two typical peaks at 1342 cm^−1^ (D peak) and 1580 cm^−1^ (G peak) were observed. The G band corresponded to the vibration of the sp^2^-hybridized carbon, whereas the D band was the consequence of the sp^3^-hybridized carbon [8]. With the increase in RGO layers, the D band becomes more and more obvious, which is caused by the rich sp^2^ carbon bond in RGO nanosheets as well as the defect-induced vibration in RGO that has many defect-rich edges after hydrothermal reduction. Moreover, Figure 2b gives the XRD patterns of the RGO/a-C:H films, indicating that all RGO/a-C:H films displayed an obvious diffraction peak at about 24° that should correspond to the (002) diffraction peak of RGO. The increased RGO layers make the (002) diffraction peaks become more and more prominent. 

### 3.2. SEM Morphological Examination

The scanning electron microscopic (SEM) surface morphologies of the a-C:H film and as-prepared RGO/a-C:H films are presented in Figure 3. The a-C:H film substrate in Figure 3a exhibits a typical uniform and smooth top-surface morphology. After the assembly of RGO sheets, all the substrate surface morphologies changed to some extent, and folds of RGO sheets were evidently observed in the whole detected areas. Moreover, with the increase in RGO layers, a more evident puckered and corrugated structure emerged on the surface. Thus, combined with the results of Raman and the XRD analysis, it was convincingly demonstrated that RGO sheets with different layers were successfully assembled on the surface of a-C:H film. 

### 3.3. Tribological Performance

The macro-friction coefficient and wear-resistance ability are the key factors for the use of RGO/a-C:H films as the solid lubrication coating in a space environment. Figure 4 exhibits the curves of friction coefficient varying with sliding time for different samples. The wear life of an unmodified a-C:H film only lasts for 480 cycles. The self-assembled RGO layers showed evident influence on the tribological property of a-C:H film, and the friction coefficient curves exhibited different trends with the increase of RGO layers. For example, when the RGO layers were 1, 7 or 10, it had no significant improvement or even exerted negative effects on the life span of a-C:H film. Only when the RGO layer was set as a moderate value of 3 or 5 layers that the RGO/a-C:H film showed the longest anti-wear lifetime of above 18,000 cycles. It is worth mentioning that the applied load was 5 N in this friction test, which is far higher than similar characterizations for self-assembled graphene in previous reports [9]. Thus, this excellent wear resistance of the RGO/a-C:H film has a direct practical value for a-C:H film to be used as a solid lubricant in a vacuum condition. The lubrication mechanism of RGO/a-C:H film still needs more in-depth investigation, but it can be reasonably inferred that the high load-carrying capacity low friction of a-C:H film and special structure of RGO sheet are certainly two main factors. In particular, graphene with a two-dimensional structure, on the one hand, is easy for the interlayer to slide under the action of shear force, on the other hand, it may also provide protection against the undesired chemical reaction of a-C:H film such as the hydrogen release process, which can, in turn, help increase the reliability of a-C:H film. Thus, under appropriate conditions (for example, a moderate graphene layer), a fantastic synergistic lubrication system can be formed between the a-C:H film and RGO layers, which plays a vital role in improving the tribological property of a-C:H film under vacuum.

## 4. Conclusions

In summary, RGO sheets with different layers on a-C:H film substrate were successfully prepared through the self-assembly method by using APTES as the intermediate coupling layer. The structural analysis and tribological characterization indicate that the as-prepared RGO/a-C:H film possesses distinctive surface morphology and more excellent anti-wear resistance ability when compared with the untreated a-C:H film. This convenient and feasible self-assembly method provides a new route for improving the tribological property of a-C:H film in a vacuum environment and demonstrates that graphitic lubricating material still has great potential in the space field application.

## Figures and Tables

**Figure 1 nanomaterials-09-01733-f001:**
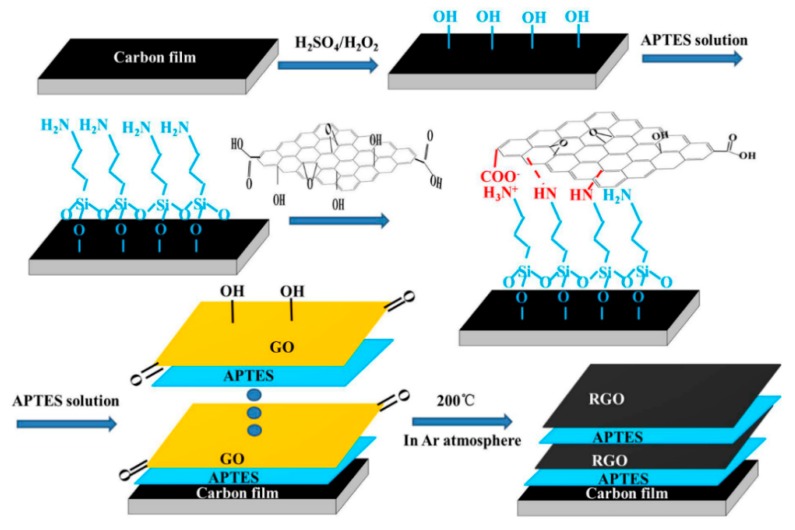
The proposed schematic view for preparing the RGO/a-C:H films.

**Figure 2 nanomaterials-09-01733-f002:**
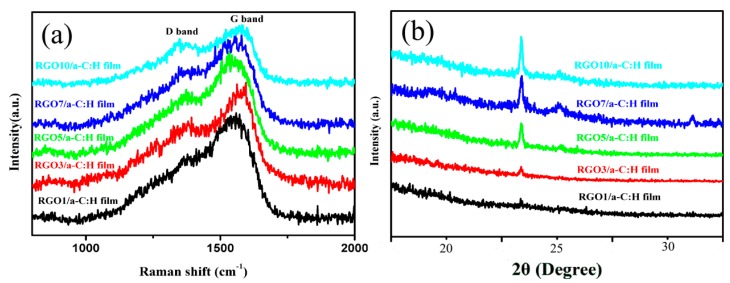
Raman spectrum (**a**) and XRD pattern (**b**) of as-prepared films.

**Figure 3 nanomaterials-09-01733-f003:**
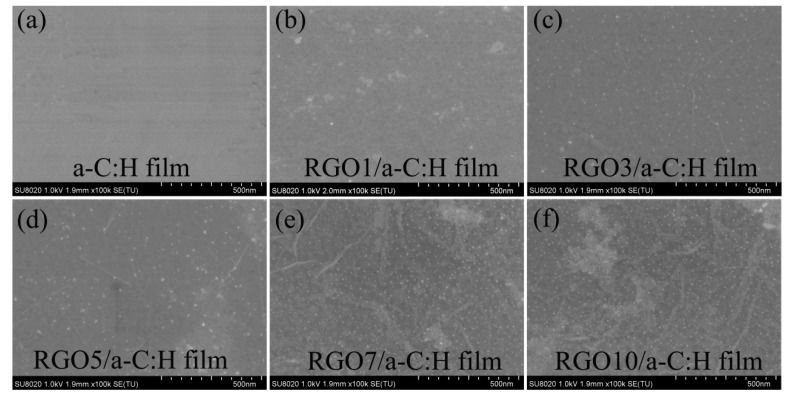
SEM images of top surfaces of the as-prepared films: (**a**) a-C:H, (**b**) RGO1/ a-C:H, (**c**) RGO3/ a-C:H, (**d**) RGO5/ a-C:H, (**e**) RGO7/ a-C:H, (**f**) RGO10/ a-C:H.

**Figure 4 nanomaterials-09-01733-f004:**
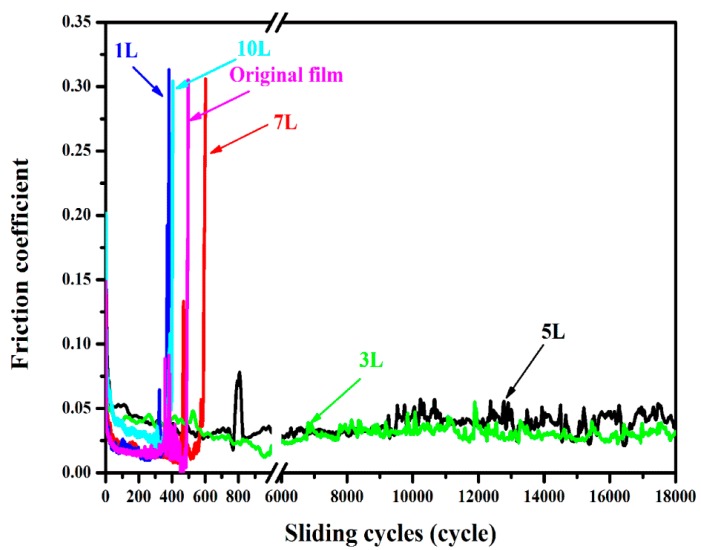
Tribological performance of the as-prepared films in a vacuum environment.

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
