# Peer review of "Improving the Wear Life of a-C:H Film in High Vacuum by Self-Assembled Reduced Graphene Oxide Layers"

_nanomaterials, 2019, doi:10.3390/nano9121733_

Round 1
Reviewer 1 Report
Please check the text carefully. There are several errors, like „hemical inrtness”, “Many literatures” – page 1, “vaule” –page 2 etc.
What was hardness of a ball?
What was a reason of selecting operating parameters?
When a limiting value of the friction coefficient?
What is the number of test repetitions?
Author Response
Reviewer #1
Please check the text carefully. There are several errors, like „hemical inrtness”, “Manyliteratures” – page 1, “vaule” –page 2 etc.
Thanks for your suggestions. We have carefully checked the manuscript and fixed such errors.
What was hardness ofa ball?
The hardness value of counterpart GCr15 ball (HRC 58-60) was given in the revised manuscript.
What was a reason of selecting operating parameters?
In this study, the operating parameters was as following: sliding speed was 300 r/min, the applied load was 5 N. These parameters is a common used parameters to characterize the tribological performance of carbon films in our previous studies and some other reports. The corresponding Hertz contact stress is as high as about 3 GPa, which is higher than most actual operating conditions. So, in our opinion, this high contact stress is high enough to evaluate the tribological performance of as-sprayed films. However, as we have preliminary investigated the effect of self-assembled reduced graphene oxide layer on tribological performance of a-C:H film in high vacuum, systematic study of tests under different loads /speed conditions were not conducted in this study. Nevertheless, for future application in space, these aspects need to be investigated and corresponding friction mechanism should be discussed.
4. When a limiting value of the friction coefficient?
If friction coefficient exceeded 0.3, the lubrication ability of the film was considered to be ineffective and the friction test was manully stopped.
5. What is the number of test repetitions?
Each friction test was conducted at least three times, the repeated tests show a similar tendency.

Reviewer 2 Report
The subject of the manuscript “Improving the wear life of a-C:H film in high vacuum by self-assembled reduced graphene oxide layers” by Hui Song et al. is focused on reduced graphene oxide (RGO) layers assembled on a-C:H film by employing APTES monolayer as the intermediate coupling agent. The friction behavior and wear mechanism of as-prepared RGO/a-C:H film were investigated in vacuum environment.
The manuscript can be accepted for publication after addressing the following issues:
“1.5 um” should read “1.5 µm” (page 1, line 51); In the “Results and discussion” section, in order for the readers to have a clear picture of the presented results, the authors are suggested to introduce 4 subsections, as following: “3.1 Raman investigations”; “3.2 XRD analyses”; “3.3 Morphological examination”; and “3.4 Tribological testing”. Some information on the roughness of the substrate and final (RGO) layer should be provided. Does this parameter have an influence on the tribological behavior of the structure? The authors should introduce an explanation in the revised manuscript. Did the authors perform a parametric study when choosing 5N as the applied load for tribological tests? What was their rationale? I find it interesting to know how does the multilayer behave when applying different loads (inferior or superior to 5N). An explanation should be introduced in the revised version of the manuscript.
Some other language and grammar issues follow:
“should be account” should read as “should be accounted” (page 1, line 17); “hardnessc” should read as “hardness” (page 1, line 24); delete one “and” (page 1, line 39); “Many literatures” should read as “Many literature studies” or simply “Many studies”; “After been” should read as “After being” (page 2, line 41); “vaule of friction” should read as “value of friction” (page 2, line 69); “manully stopped” should read “manually stopped” (page 2, line 70); “the D band become” should read as “the D band becomes” (page 2, line 77); “Fig. 2(b) give XRD” should read as “Fig. 2(b) gives XRD” (page 2, line 79); “should be corresponded” should read “should correspond” (page 2, line 81); “can also provides” should read as “can also provide” (page 3, line 105); “moderate grapheme” should read as “moderate graphene” (page 3, line 108); “more excellent” should read as “improved” or simply “excellent” (page 3, line ).
Author Response
The subject of the manuscript “Improving the wear life of a-C:H film in high vacuum by self-assembled reduced graphene oxide layers” by Hui Song et al. is focused on reduced graphene oxide (RGO) layers assembled on a-C:H film by employing APTES monolayer as the intermediate coupling agent. The friction behavior and wear mechanism of as-prepared RGO/a-C:H film were investigated in vacuum environment. The manuscript can be accepted for publication after addressing the following issues:
First, we would like to express our appreciation for your impersonal comments and helpful suggestions on our work. We have carefully considered these questions and have made detailed revisions as best as we could.
1. “1.5 um” should read “1.5 µm” (page 1, line 51)
Thanks for reminding, we have modified the corresponding content in the revised manuscript.
2. In the “Results and discussion” section, in order for the readers to have a clear picture of the presented results, the authors are suggested to introduce 4 subsections, as following: “3.1 Raman investigations”; “3.2 XRD analyses”; “3.3 Morphological examination”; and “3.4 Tribological testing”.
Thanks for your suggestion. According to your suggestion, the “Results and discussio” section is divided into 3 subsections: “3.1 Raman and XRD characterization”, “3.2 SEM morphological examination” and “3.3 Tribological performance”.
3. Some information on the roughness of the substrate and final (RGO) layer should be provided. Does this parameter have an influence on the tribological behavior of the structure?
Original carbon film 1layer of RGO 5 layer of RGO
Thanks for your suggestion. There are indeed some differences in roughness of original carbon film and RGO coated film. For example, from the SEM images in manuscript and the following AFM images of test samples, it can be found the difference of roughness. The original carbon film is very smooth and the roughness is only 1.5nm. The self-assembled RGO to some extent increase the roughness. Just as you mentioned, roughness may has some impacts on tribological performance. However, as tribological performance is related to many factors, the influence of roughness needs a deep investigation in future study.
4. The authors should introduce an explanation in the revised manuscript. Did the authors perform a parametric study when choosing 5N as the applied load for tribological tests? What was their rationale? I find it interesting to know how does the multilayer behave when applying different loads (inferior or superior to 5N). An explanation should be introduced in the revised version of the manuscript.
Thanks for your comments. 5N is a common used applied load in ball-on-disk tribometer, which is often used to characterize the tribological property of amorphous cabon films in our previous study and some other reports. In this study, as we only preliminary investigated the effect of self-assembled reduced graphene oxide layer on tribological performance of a-C:H film in high vacuum, only a single applied load of 5N was used. However, just as you said, the film may exhibit different tribological performance under different loads. This aspect should be further investigated in the following studies.

Round 2
Reviewer 2 Report
The authors took into consideration most of the raised queries. Unfortunately, there are still some grammar errors which were noticed by this Reviewer and not taken into consideration by the authors and therefore they should be solved with the Proof.